# Factors associated with the uptake of intermittent preventive treatment for malaria during pregnancy in Cameroon: An analysis of data from the 2018 Cameroon Demographic and Health Survey

**Dominique Ken Guimsop** *, **Ange Faustine Kenmogne Talla**, **Haoua Kodji**, **Jerome Ateudjieu**

Departement of Public Health, Faculty of Medicine and Pharmaceutical Sciences, University of Dschang, Dschang, Cameroon

* dk.guimsop@outlook.com

⌕ OPEN ACCESS

**Data Availability Statement:** https://doi.org/10. 6084/m9.figshare.21308949.

## Abstract

Malaria in pregnancy is a major public health concern that contributes to a significant increase in maternal and child mortality and morbidity. Intermittent preventive treatment of malaria during pregnancy using sulfadoxine-pyrimethamine (IPTp-SP) is a key intervention recommended by the World Health Organization (WHO) and implemented in Cameroon to reduce the morbidity associated with malaria during pregnancy. This study aimed to assess the distribution of the poor uptake of IPTp-SP (i.e. fewer than three doses) in Cameroon and the factors associated. We conducted a secondary analysis of data extracted from the 2018 Cameroon Demographic and Health Survey. Data was collected using a face-to-face questionnaire administered to mothers with at least one child under the age of five. The participants were selected using a two-stage stratified sampling process. We estimated the frequencies of mothers receiving fewer than three doses of IPTp-SP. Multilevel logistic regression modeling was used to assess the associations between key suspected determinants and uptake of fewer than three doses of IPTp-SP. Crude and adjusted Odds-Ratio (ORs) were estimated. A total of 13,527 women of childbearing age were interviewed, of whom 5,528 (40.9%) met our selection criteria. Among them, 845 (15.3%) women had no antenatal consultation (ANC) visit, 1,109 (20%) had 1–3 visits, 3,379 (61.1%) had 4–7 visits, and only 195 (3.5%) had at least eight visits. Moreover, 3,398 (61.5%, CI: 60.2–62.8) had received fewer than three doses of IPTp-SP. Our findings show that the predictors of poor uptake of IPTp-SP include attending the first ANC visit after the third month of pregnancy (aOR = 1.52, CI: 1.30–1.77), attending fewer than four ANC visits (aOR = 1.29, CI: 1.06–1.56), and not being attended to by a healthcare professional during the prenatal period (aOR = 4.63, CI: 2.81–7.64). Residing in the Sahelian regions was not increasing the risk of poor IPTp-SP uptake on its own but was positively modifying the effect of not being attended by a healthcare professional (p < 0.001). We did not find a significant association between a higher level of education and the uptake of IPTp-SP (aOR = 1.10, CI: 0.90–1.32). Nearly two

**Funding:** The authors received no specific funding for this work.

**Competing interests:** The authors have declared that no competing interests exist.

third of the pregnant women in Cameroon have a poor uptake of IPTp-SP. Interventions focused on ANC provision ought to be explored and tested to address this gap, with priority assigned to the Sahelian region.

## Introduction

Malaria, caused in humans by five species of *Plasmodium*, remains one of the most widespread infectious diseases. The disease is primarily caused by *Plasmodium falciparum* and *Plasmodium vivax*, and poses a threat to nearly half of the global population. In 2019, 229 million cases of malaria were reported, resulting in 409,000 deaths, with over 90% occurring in Africa [1]. Malaria is known to significantly increase the risk of maternal and fetal anemia, abortion, low birth weight, and neonatal death [2–5]. To mitigate the impact of this disease, efforts have focused on three main strategies: intermittent preventive treatment during pregnancy using sulfadoxine-pyrimethamine (IPTp-SP), targeted use of insecticide-treated nets, and effective case management in areas with moderate to severe transmission [6]. The IPTp-SP policy recommends that each pregnant woman receive a minimum of three doses of sulfadoxine-pyrimethamine (SP) during her pregnancy, with each dose safely administered at least one month apart, starting at 13 weeks of gestation and lasting till delivery [7, 8].

Despite the implementation of proper interventions, the overall administration of the recommended three doses of IPTp-SP was only 34% in 2020 among the 33 countries that had adopted this preventive measure [1]. Several factors contribute to this poor rate, including limited access to healthcare, inadequate training of healthcare providers, drug stock-outs, and maternal and pregnancy-related factors such as the level of education and multigravidity [9–12].

In Cameroon, the Ministry of Health adopted IPTp in May 2002 through the administration of chloroquine as part of antenatal consultation (ANC) visits. Subsequently, in January 2004, IPTp using chloroquine was replaced by IPTp-SP following WHO recommendations [13]. Existing literature estimates that the recommended three-dose uptake coverage of IPTp-SP in Cameroon ranges from 23.5% to 54.9%, depending on socio-demographic characteristics, ANC performance, and study design [9–14]. These figures fall significantly short of the country's target coverage rate of 80% by 2023. However, there is limited literature available, from a nationally representative sample, which have explored socio-demographic and antenatal parameters that may influence the uptake of IPTp-SP in Cameroon. The range of investigated factors was non-exhaustive, and we observe some mitigated results that are attributable to the use of unharmonized procedures and varying study designs. Due to the potential value of such findings in informing policymakers about strategies to improve the effectiveness of existing approaches or proposing new ones, we seek to contribute to the understanding of the mechanisms responsible for the poor uptake of IPTp-SP. Therefore, this study aims to evaluate the distribution of poor uptake of IPTp-SP among pregnant women in Cameroon and assess its determinants.

## Methodology

### Study design and setting

The Demographic and Health Survey (DHS) program conducts nationally representative household surveys in over 90 countries and provides indicators on a wide range of topics, including population, wealth, maternal and child health, fertility and family planning, nutrition, and education. The DHS sampling is implemented using a two (or sometimes a three)-stage stratified sampling design using censuses as sampling frames.

For this study, we conducted a secondary data analysis of the 2018 Cameroon Demographic and Health Survey [14]. The data was collected from a cross-sectional household-based survey targeting 22 territorial units in Cameroon, including the 10 regions (each divided into their respective urban and rural areas) and the two major cities in the country. A unit affected by the ongoing socio-political crisis was excluded. The households were selected using a stratified cluster random sampling process carried out in two stages. In the first stage, 470 standard enumeration areas (SEAs), also known as clusters (245 SEAs in urban areas and 225 in rural areas), were selected with probability proportional to the SEA size. In the second stage, a fixed number of 28 households per cluster were selected through equal probability systematic sampling from each cluster's household listing. Data was collected from each visited household using a face-to-face questionnaire. A total of 11,710 households were surveyed, from which 13,527 women of childbearing age completed the interview. A detailed methodology can be found in the DHS final report [14].

## Study population

Our target population included all women aged 15–49 years who resided in the selected cluster and were biological mothers of children under five years of age on the day the selected household was visited.

## Participants' data extraction from the database

The DHS uses standardized questionnaires, including those for households, women, men, and biomarkers, to collect information on household and individual characteristics. Optional modules can be added or adapted to meet country-specific needs. The DHS data undergoes thorough cleaning and anonymization, and the datasets are made publicly available to researchers upon request. For our study, we extracted cases from the CMRDHS women's dataset, which was collected using the women's questionnaire [15]. Information regarding the variable "area's malaria prevalence" was obtained from the 2018 DHS geospatial covariate database [16] and incorporated into our dataset.

To identify our participants, we derived the variable "mother of an under-five-year-old child" from the variable "woman of childbearing age" through coding. From the 13,527 women of childbearing age who completed the interview, we selected the 6,463 who had a live birth in the five years preceding the survey. Among them, 1054 women were excluded from the analysis, as their responses had missing data on any of our key variables (attending ANCs and taking SP during pregnancy). Ultimately, the sample for this study comprised 5,409 cases (which increased to 5,528 after applying the weighting factors), from which we captured the responses (Fig 1).

## Variable of interest

For the analysis, we extracted data based on variables that were theoretically and empirically linked to the uptake of three or more doses of IPTp-SP:

(a) The dependent variable was the uptake of IPTp-SP, which comprised two categories: fewer than three doses (also referred to as "poor uptake") and three or more doses.

(b) The independent variables consisted of both individual-level variables and cluster (or community)-level variables. The individual-level variables included a woman's age, length of stay in the place of residence, level of education, religion, literacy (ability to read a written phrase in the language of choice), wealth (categorized by DHS into quintiles, numbered 1–5 in order of increasing wealth, based on a composite measure of the household cumulative living standard) [17], health insurance coverage, occupational status, marital status, partner's level of

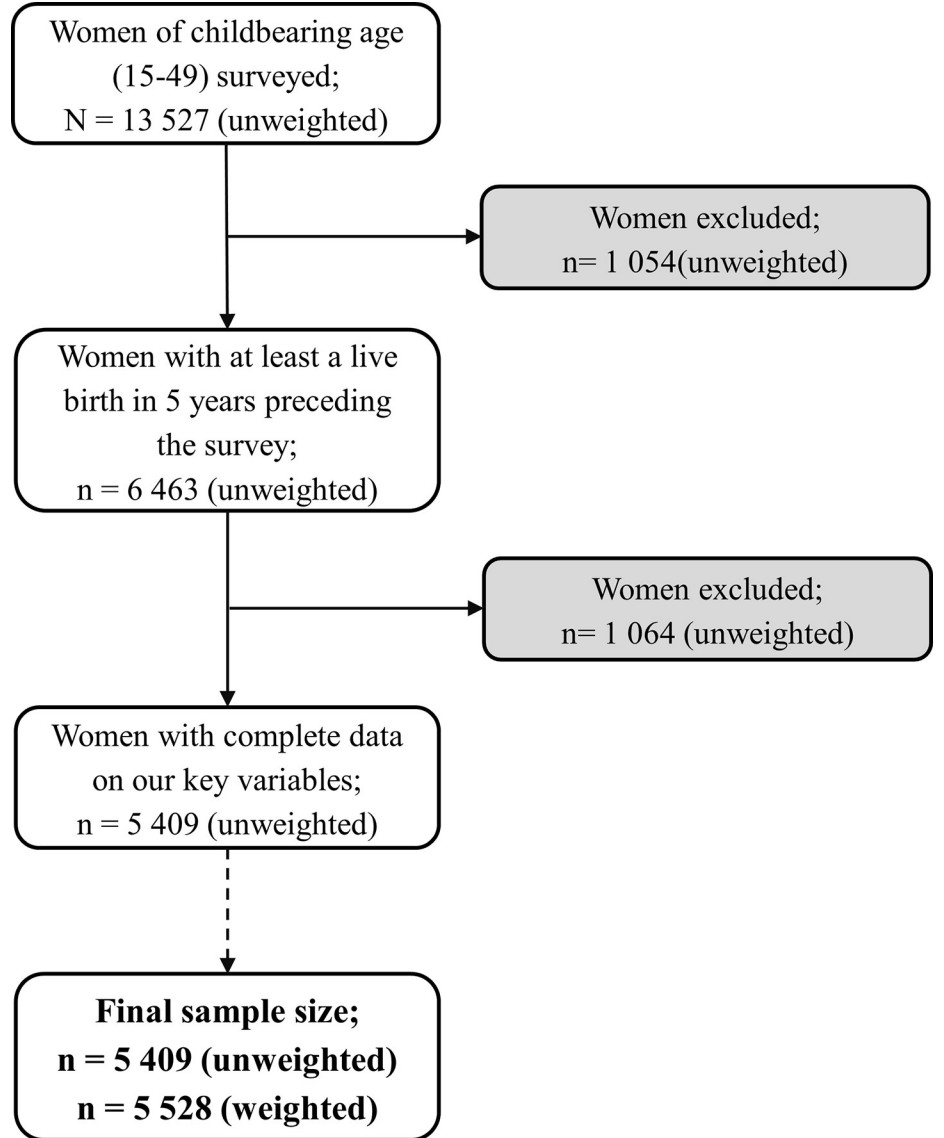

**Fig 1. Flow diagram of sample selection from the 2018 Cameroon Demographic and Health Survey data.**

education, desire for pregnancy, parity, history of pregnancy termination, qualification of the ANC attendant, and number of ANC visits. The community-level variables included region (including the 10 official regions and separate categories for the two largest urban areas, Douala and Yaoundé, resulting in 12 regional categories), area of residence (rural, defined as a population density of < 20,000 people), and the area's malaria prevalence. Variables pertaining to pregnancy and prenatal care were specific to the most recent live birth.

## Data management

Data extraction, cleaning, and analysis were carried out using IBM SPSS Statistics version 26 (International Business Machines Corp., New York, USA). The sampling design of the survey was accounted for by using the "Complex Samples" package of IBM SPSS for the analyses. This package allows for the specification of the sample scheme parameters, including

stratification, clustering, weighing, and inclusion probabilities, in order to accurately estimate standard errors. Sample weights were provided by the "weight" variable from the database. Unless otherwise specified, only weighted data was presented in the results.

## Statistical analysis

Categorical variables were summarized as frequencies and proportions. The Rao-Scott chi-square test was used to compare the distribution of the independent variables between the two levels of our dependent variable. This analysis tested for differences within groups while adjusting for sample design using an F statistic and was done using independents categorical variables in their initial (non-dichotomized) form (Tables 1 and 2). All independent variables were subsequently dichotomized, and bivariate logistic regression was used within the multi-level modelling framework to measure the independent variables' association with the uptake of fewer than three doses of IPTp-SP (Table 3). The independent variables that were associated with the outcome at a significance level of $p < 0.20$ were selected as candidates for the main effects in the multilevel binary logistic regression model. Two independent variables ("length of stay in the place of residence" and "current occupation") did not reach that significance level in the bivariate regression analysis and were, therefore, excluded from the multilevel model. Collinearity was checked by calculating the variance inflation factors (VIF). Covariates with a VIF greater than 4, indicating a twofold increase in the standard error of a regression coefficient in the presence of collinearity, were considered to have high collinearity. The variable "illiteracy" was removed from the model due to suspected collinearity with "level of education."

Next, the two-level (multilevel) binary logistic regression analysis was performed to examine the association (measure of effect) between the outcome and each independent variable while controlling for the effects of the other variables. This model estimated the adjusted odds ratio (aOR) ($\exp(\beta)$) while accounting for the unobserved characteristics at the community level during the data collection in addition to the women's individual characteristics. The notation of the model is as follows:

$$log\left(\frac{\pi_{ij}}{1 - \pi_{ij}}\right) = \beta_0 + \beta_1 x_{1ij} + \beta_2 x_{2j} + u_j$$

where, $u_j \sim N(0, \sigma_u^2)$ and $\pi_{ij}$ = the probability of an event occurring for the $i$ level-1 unit in the $j$ level-2 unit; $\beta_0$ is the log-odds that $y = 1$ when $x = 0$ and $u = 0$; $\beta_1$ is the effect on log-odds of a 1-unit increase in $x$ for individuals in the same group; $u_j$ is the effect of being in group $j$ on the log-odds that $y = 1$, also known as a level-2 residual; $\sigma_u^2$ is the level-2 (residual) variance or the between-group variance in the log-odds that y = 1 after accounting for x; $x_{1ij}$ is a generic level 1 nested within a level-2 independent variable; and $x_{2j}$ indicates a level-2 independent variable.

The design effect, which represents the ratio of the variance of the estimate to the variance obtained by assuming that the sample is a simple random one, was first checked on an intercept-only model. We found a significant design effect of 128.01, indicating the substantial impact of accounting for the complex design. Random intercepts were retained. The full model was then ran by entering all the selected independent variables simultaneously.

Substantive cross-level interactions among our independent variables were explored through a trial-and-error process to capture the potential effect of community-level characteristics on the individual-level associations. Only one interaction was found to be significant and was consequently retained in the model. The final model achieved a Nagelkerke $R^2$ of 0.201, indicating a moderate level of explanatory power. The level of significance used was 5% (0.05), two-tailed at the 95% confidence interval (CI).

**Table 1. Socio-demographic characteristics and comparison between those who took fewer than three doses of IPTp-SP and the others (N = 5,528).**

| Variable | Total | IPTp-SP uptake, n (row %) | | Rao-Scott F(df1, df2) | p |
|---|---|---|---|---|---|
| | n (col %) N = 5528 | < 3 doses n = 3398 | 3 + doses n = 2130 | | |
| Woman's age (years) | | | | | |
| 15–19 | 521 (9.4) | 333 (63.9) | 188 (36.1) | 0.93 (4.53, 1846.71). | 0.457 |
| 20–24 | 1205 (21.8) | 760 (63.1) | 445 (36.9) | | |
| 25–29 | 1496 (27.1) | 918 (61.4) | 578 (38.6) | | |
| 30–34 | 1161 (21.0) | 703 (60.6) | 459 (39.5) | | |
| 35–39 | 727 (13.1) | 421 (57.9) | 305 (42) | | |
| 40–49 | 416 (7.5) | 262 (63) | 155 (37.3) | | |
| Length of stay in the place of residence | | | | | |
| Visitor | 121 (2.2) | 79 (65.3) | 42 (34.7) | 0.26 (2.83, 1158.50) | 0.844 |
| 0–19 years | 3454 (62.5) | 2119 (61.3) | 1334 (38.6) | | |
| 20–40 years | 219 (4.0) | 140 (63.9) | 79 (36.1) | | |
| Always | 1734 (31.4) | 1060 (61.1) | 674 (38.9) | | |
| Level of education | | | | | |
| None | 1425 (25.8) | 996 (69.9) | 429 (30.1) | 16.24 (2.77, 1130.29) | <0.001 |
| Primary | 1647 (29.8) | 1058 (64.2) | 590 (35.8) | | |
| Secondary | 2121 (38.4) | 1185 (55.9) | 936 (44.1) | | |
| Higher | 334 (6.1) | 160 (47.9) | 175 (52.4) | | |
| Religion | | | | | |
| Animism/Other | 212 (3.8) | 145 (68.4) | 67 (31.6) | 4.70 (1.97, 805.91) | 0.010 |
| Christianity | 3786 (68.5) | 2246 (59.3) | 1540 (40.7) | | |
| Islam | 1530 (27.7) | 1007 (65.8) | 523 (34.2) | | |
| Illiteracy | 2016 (36.5) | 1391 (69) | 625 (31) | 30.88 (1.0, 408.0) | <0.001 |
| Combined wealth index | | | | | |
| Poorest | 1111 (20.1) | 779 (70.1) | 331 (29.8) | 22.38 (3.75, 1531.02) | <0.001 |
| Poorer | 1215 (22) | 842 (69.3) | 374 (30.8) | | |
| Middle | 1135 (20.5) | 734 (64.7) | 401 (35.3) | | |
| Richer | 1100 (19.9) | 587 (53.4) | 514 (46.7) | | |
| Richest | 967 (17.5) | 457 (47.3) | 510 (52.7) | | |
| Health insurance coverage | 99 (1.8) | 45 (45.5) | 54 (54.5) | 7.62 (1.0, 408.0) | 0.006 |
| Current occupation | 3769 (68.3) | 2306 (61.2) | 1463 (38.8) | 0.21 (1.0, 408.0) | 0.650 |
| Union involvement at any point | 4448 (80.5) | 2767 (62.2) | 1681 (37.8) | 3.74 (1.0, 408.0) | 0.054 |
| Partner's level of education, secondary and above | 3636 (65.8) | 2336 (64.2) | 1299 (35.7) | 21.87 (1.0, 408.0) | <0.001 |
| Region | | | | | |
| Sahelian region | 2025 (36.6) | 1326 (65.5) | 699 (34.5) | 20.0 (1.94, 793.42) | <0.001 |
| Southern region | 2528 (45.7) | 1611 (63.7) | 917 (36.3) | | |
| Douala/Yaoundé | 975 (17.6) | 461 (47.3) | 515 (52.8) | | |
| Residing in a rural area | 2911 (52.6) | 2000 (68.7) | 911 (31.3) | 48.75 (1, 41) | <0.001 |
| Area's malaria prevalence (%) | | | | | |
| 0–7 | 594 (10.7) | 76 (12.8) | 518 (87.2) | 26.64 (4.11, 1677.85) | 0.000 |
| 8–15 | 1165 (21.1) | 195 (16.7) | 970 (83.3) | | |
| 16–23 | 1645 (29.8) | 695 (42.2) | 950 (57.8) | | |
| 24–31 | 1041 (18.8) | 417 (40) | 624 (60) | | |
| 32–40 | 818 (14.8) | 440 (53.8) | 378 (46.2) | | |
| 41+ | 265 (4.8) | 131 (49.5) | 134 (50.5) | | |

Intermittent preventive treatment of malaria during pregnancy using sulfadoxine-pyrimethamine (IPTp-SP)

Note: all presented sample sizes and frequencies are weighted

**Table 2. Obstetrical history and comparison between those who took fewer than three doses of IPTp-SP and the others (N = 5,528).**

| Variable | Total | IPTp-SP uptake, n (row %) | | Rao-Scott F(df1, df2) | p |
|---|---|---|---|---|---|
| | n (col %) N = 5528 | < 3 doses n = 3398 | 3+ doses n = 2130 | | |
| Desired pregnancy | 5280 (95.6) | 3244 (61.4) | 2036 (38.6) | 0.02 (1, 408) | 0.886 |
| Parity ≤ 5 | 4432 (80.2) | 2682 (60.5) | 1750 (39.5) | 4.56 (1, 408) | 0.033 |
| Event of a prematurely terminated pregnancy | 1023 (18.5) | 584 (57.1) | 439 (42.9) | 5.29 (1, 408) | 0.022 |
| Qualification of the ANC attendant | | | | | |
| No care | 845 (15.3) | 816 (96.6) | 29 (3.4) | 88.04 (3.21, 1310.16) | <0.001 |
| Doctor | 1495 (27) | 753 (50.4) | 742 (49.6) | | |
| Nurse/midwife | 3012 (54.5) | 1734 (57.6) | 1278 (42.4) | | |
| Auxiliary midwife | 169 (3.1) | 89 (52.7) | 80 (47.3) | | |
| Others* | 7 (0.1) | 6 (85.7) | 1 (14.3) | | |
| First ANC visit after the third month | 3253 (58.9) | 2283 (70.2) | 970 (29.8) | 172.31 (1, 408) | <0.001 |
| Number of ANCs | | | | | |
| 0 | 845 (15.3) | 816 (96.6) | 29 (3.4) | 67.40 (4.61, 1880.81) | <0.001 |
| 1 | 83 (1.5) | 59 (71.1) | 24 (28.9) | | |
| 2 | 233 (4.2) | 167 (71.7) | 66 (28.3) | | |
| 3 | 793 (14.3) | 469 (59.1) | 324 (40.9) | | |
| 4–7 | 3379 (61.1) | 1792 (53) | 1586 (46.9) | | |
| 8 + | 195 (3.5) | 95 (48.7) | 100 (51.3) | | |

Intermittent preventive treatment of malaria during pregnancy using sulfadoxine-pyrimethamine (IPTp-SP); antenatal consultation (ANC)

*"Others" includes traditional birth attendants and community/village worker; Note: all presented sample sizes and frequencies are weighted

## Ethical consideration

The Cameroon DHS protocol, which included measurement procedures and biological tests, underwent review, and obtained approval from the Cameroonian Ministry of Health, the National Ethical Committee for Research and Human Health (CNERSH), and the Inner-City Fund (ICF) Institutional Review Board. Informed consent for the survey was obtained from each participant before their interview. Permission to access the Cameroon DHS 2018 survey dataset was obtained from the DHS program. The surveyors ensured confidentiality by using an anonymization technique that assigned a code to each included case. The accessed dataset was securely stored on a computer protected by password authentication.

## Results

### Socio-demographic characteristics of the study population and comparison with the uptake of IPTp-SP

We included 5,528 participants from the database. Nearly half of them (48.1%, n = 2,657) were aged between 25 and 35 years, and 44.5% (n = 2,455) had obtained a secondary school education at least. More than half of the women (52.6%, n = 2,911) resided in rural areas, and there was a relatively even distribution across different wealth classes. Finally, 3,398 (61.5%, CI: 60.2–62.8) participants had taken fewer than three doses of IPTp-SP (Table 1).

Up to 845 (15.3%) women did not attend any ANCs. Among the 3,574 (64.6%) women who attended the recommended number of ANCs, 1,686 (47.2%) received three or more doses of IPTp-SP (Fig 2).

Table 1 presents comparative analyses to show no significant association between the length of stay in the place of residence and the uptake of IPTp-SP (p = 0.844). However, we observe a

**Table 3. Factors associated with the uptake of fewer than three doses of IPTp-SP.**

| Variable | IPTp-SP uptake, n (row %) | | Bivariate analysis | | Multivariate analysis | |
|---|---|---|---|---|---|---|
| | < 3 doses n = 3398 | 3 + doses n = 2130 | OR (95% CI) | p | aOR (95% CI) | p |
| Individual-level characteristic | | | | | | |
| Level of education | | | | | | |
| None/primary | 2053 (66.8) | 1019 (43.2) | 1.66 (1.43–1.93) | < 0.001 | 1.08 (0.89–1.32) | 0.425 |
| Secondary and higher | 1345 (54.8) | 1111 (65.2) | Ref. | | | |
| Christian | | | | | | |
| Yes | 2246 (59.3) | 1540 (40.7) | 0.74 (0.61–0.91) | 0.004 | 1.00 (0.79–1.29) | 0.957 |
| No | 1152 (66.1) | 590 (33.9) | Ref. | | | |
| Combined wealth index | | | | | | |
| Poor | 1621 (69.7) | 705 (30.1) | 1.84 (1.54–2.20) | < 0.001 | 1.08 (0.87–1.33) | 0.500 |
| Middle to rich | 1777 (55.5) | 1425 (44.5) | Ref. | | | |
| Health insurance coverage | | | | | | |
| Yes | 45 (45.5) | 54 (54.5) | 0.51 (0.32–0.83) | 0.007 | 0.75 (0.47–1.20) | 0.230 |
| No | 3353 (61.8) | 2076 (38.2) | Ref. | | | |
| Union involvement at any point | | | | | | |
| Yes | 2767 (62.2) | 1681 (37.8) | 1.17 (0.99–1.37) | 0.054 | 1.04 (0.85–1.28) | 0.682 |
| No | 631 (58.4) | 449 (41.6) | Ref. | | | |
| Partner's level of education: secondary and above | | | | | | |
| Yes | 1062 (56.1) | 831 (43.9) | 1.41 (1.21–1.62) | < 0.001 | 0.92 (0.77–1.10) | 0.377 |
| No | 2336 (64.3) | 1299 (35.7) | Ref. | | | |
| Desired pregnancy | | | | | | |
| Yes | 3244 (61.4) | 2036 (38.6) | 0.89 (0.74–1.05) | 0.182 | 1.01 (0.71–1.45) | 0.950 |
| No | 154 (62) | 94 (38) | Ref. | | | |
| Parity $\leq$ 5 | | | | | | |
| Yes | 2682 (60.5) | 1750 (39.5) | 0.81 (0.67–0.98) | 0.033 | 1.02 (0.82–1.26) | 0.861 |
| No | 716 (65.3) | 380 (34.7) | Ref. | | | |
| Prematurely terminated pregnancy | | | | | | |
| Yes | 584 (57.1) | 439 (42.9) | 0.80 (0.66–0.96) | 0.022 | 0.88 (0.72–1.09) | 0.255 |
| No | 2814 (62.5) | 1691 (37.5) | Ref. | | | |
| ANC from a healthcare professional | | | | | | |
| No | 822 (96.5) | 30 (3.5) | 4.59 (2.81–7.51) | < 0.001 | 4.63 (2.81–7.64) | **< 0.001** |
| Yes | 2576 (55.1) | 2100 (44.9) | Ref. | | | |
| First ANC visit after the third month | | | | | | |
| Yes | 2283 (70.2) | 970 (29.8) | 2.45 (2.14–2.80) | < 0.001 | 1.52 (1.30–1.77) | **< 0.001** |
| No | 1115 (49) | 1160 (51) | Ref. | | | |
| Number of ANCs | | | | | | |
| < 4 | 1511 (77.3) | 443 (22.7) | 3.05 (2.53–3.66) | < 0.001 | 1.29 (1.06–1.56) | **0.009** |
| $\geq$ 4 | 1887 (52.8) | 1687 (47.2) | Ref. | | | |
| Community-level characteristic | | | | | | |
| Region | | | | | | |
| Sahelian regions | 1326 (65.5) | 699 (34.5) | 2.12 (1.66–2.70) | < 0.001 | 0.52 (0.39–0.68) | **< 0.001** |
| Douala/Yaoundé | 461 (47.2) | 515 (52.8) | 1.96 (1.57–2.45) | < 0.001 | 0.65 (0.51–0.85) | **0.001** |
| Southern regions | 1611 (63.7) | 917 (36.3) | Ref. | | | |
| Type of place of residence | | | | | | |
| Rural | 2000 (68.7) | 911 (31.3) | 1.91 (1.59–2.30) | < 0.001 | 1.25 (0.97–1.63) | 0.088 |
| Urban | 1398 (53.4) | 1219 (46.6) | Ref. | | | |
| Malaria prevalence (%) | | | | | | |

*(Continued)*

**Table 3.** (Continued)

| Variable | IPTp-SP uptake, n (row %) | | Bivariate analysis | | Multivariate analysis | |
|---|---|---|---|---|---|---|
| | < 3 doses n = 3398 | 3 + doses n = 2130 | OR (95% CI) | p | aOR (95% CI) | p |
| 0–7 | 76 (12.8) | 518 (87.2) | Ref. | | | |
| 8–15 | 195 (16.7) | 970 (83.3) | 1.03 (0.77–1.38) | 0.837 | 1.16 (0.87–1.54) | 0.312 |
| 16–23 | 695 (42.2) | 950 (57.8) | 1.44 (1.11–1.88) | 0.006 | 1.20 (0.90–1.60) | 0.221 |
| 24–31 | 417 (40) | 624 (60) | 1.36 (1.04–1.78) | 0.027 | 1.05 (0.45–1.46) | 0.792 |
| 32–40 | 440 (53.8) | 378 (46.2) | 1.52 (1.15–2.01) | 0.003 | 0.83 (0.60–1.53) | 0.265 |
| 40 + | 131 (49.5) | 134 (50.5) | 2.05 (1.27–3.30) | 0.003 | 1.15 (0.64–2.06) | 0.631 |
| Interaction term | | | | | | |
| ANC from a healthcare professional *by* Sahelian regions* | 1324 (63.7) | 755 (36.3) | −2.263 (0.457) | <0.001 | −2.345 (0.468) | **<0.001** |

Intermittent preventive treatment of malaria during pregnancy using sulfadoxine-pyrimethamine (IPTp-SP); Antenatal Care (ANC); Reference (ref.)

* Interaction effect (s.e); Note: all presented sample sizes and frequencies are weighted

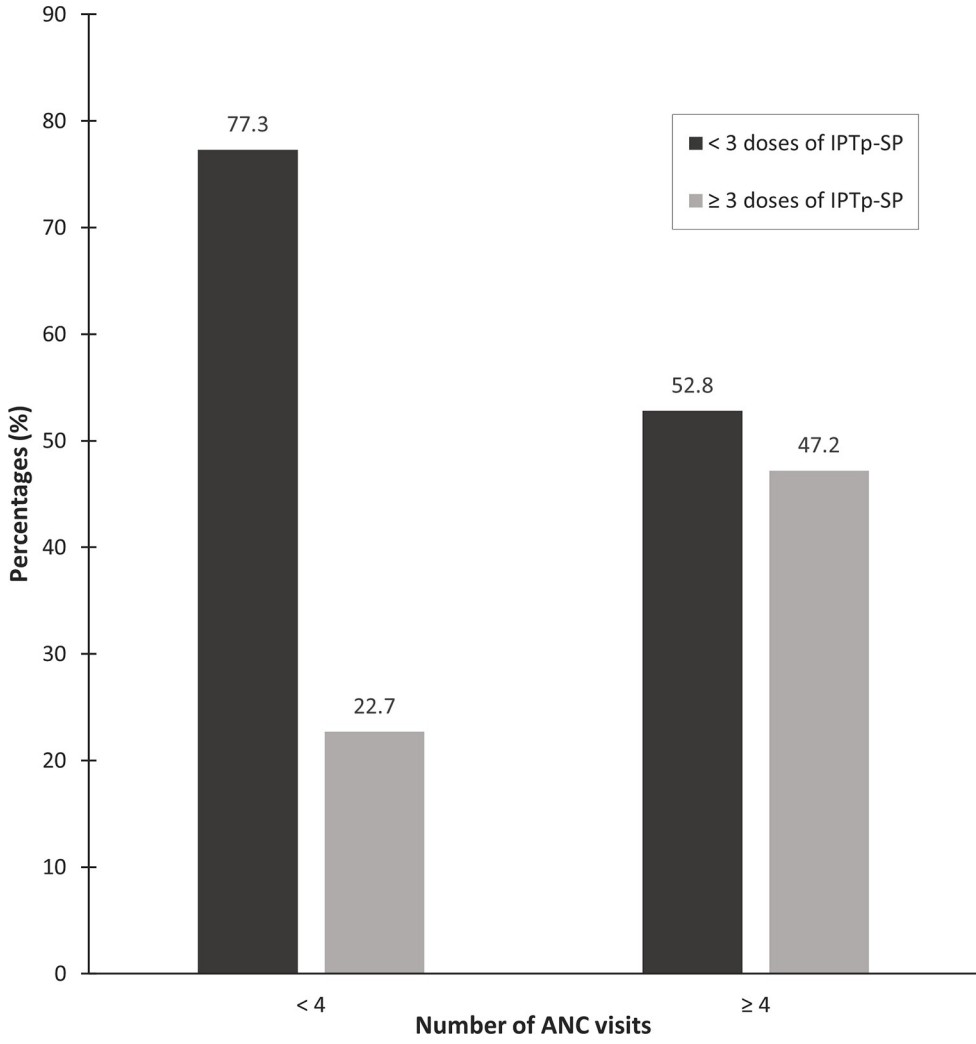

**Fig 2. Uptake of fewer than three doses of IPTp-SP according to the number of ANC visits.**

significant variation in the uptake of the recommended three doses of IPTp-SP across the different levels of education (p < 0.001). Women who had higher levels of education were more likely to take three or more doses. Illiterate participants had a significantly lower uptake of IPTp-SP compared to their literate counterparts (31% vs. 69%, p < 0.001). Health insurance coverage was limited (1.8%, n = 99); however, among the insured participants, a greater proportion took three or more doses of IPTp-SP (p = 0.006). Women who had ever been in a formal union had a slightly lower uptake of IPTp-SP compared to those who had not (62.2% vs. 37.8%), but this difference was not statistically significant (p = 0.054). Women whose partners had secondary or higher education had a lower uptake of IPTp-SP compared to those whose partners had lower levels of education (35.7% vs. 64.2%, p < 0.001). Women residing in the Sahelian regions had a significantly lower uptake of IPTp-SP compared to those living in other regions (50.5% vs. 65.5%, p < 0.001). The uptake of IPTp-SP was also found to be significantly associated with the distribution of women across areas with different levels of endemicity (p = 0.002) (Table 1).

## Obstetrical history of the study population and comparison with the uptake of IPTp-SP

Table 2 presents data on the obstetrical history of the participants. Women with a parity ≤ 5 had a lower uptake of IPTp-SP than those with a higher parity (p = 0.033). Similarly, women with a history of terminated pregnancies also had a lower uptake of IPTp-SP (p = 0.022). There was a significant association between the qualification of the ANC attendant and the uptake of IPTp-SP (p < 0.001). Among the participants who did not attend a single ANC, 96.6% had poor uptake of IPTp-SP. The majority of women (58.9%) did not attend their first ANC until their fourth month of pregnancy; this group had a generally poor uptake of IPTp-SP (p < 0.001). Most women (61.1%) had 4–7 ANCs. Additionally, the proportion of women with poor uptake of IPTp-SP progressively decreased with the number of ANCs (p < 0.001) (Table 2).

## Factors associated with the uptake of IPTp-SP

The logistic regression analyses presented in Table 3 identified independent predictors of the poor uptake of IPTp-SP. Women residing in areas of lower endemicity had a higher risk of receiving fewer than three doses of IPTp-SP (aOR = 1.26, p = 0.047), while those residing in the Sahelian regions (aOR = 0.52, p < 0.001) had a lower risk. Delay in the first ANC after the third month of pregnancy (aOR = 1.52, p < 0.001) also increased the risk of poor uptake of IPTp-SP. Similarly, attending fewer than four ANCs (aOR = 1.29, p = 0.009) and not being attended to by a healthcare professional (comprising medical doctors, nurses, and midwives) during the prenatal period (aOR = 4.63, p < 0.001) were identified as independent risk factors for the poor uptake of IPTp-SP. Additionally, the model revealed a significant positive cross-level interaction effect between not being attended to by a healthcare professional during the prenatal period and residing in the Sahelian regions (p < 0.001) (Fig 3). We found no significant association between an area's malaria prevalence and poor IPTp-SP uptake.

## Discussion

In Cameroon, IPTp-SP has been a critical strategy for preventing malaria among pregnant women since 2004. As efforts continue, a better understanding of the parameters of this service utilization can help improve its performance and inform the implementation of novel strategies. This study aimed to examine the current levels and factors associated with the uptake of the recommended three or more doses of IPTp-SP among pregnant women in Cameroon. We

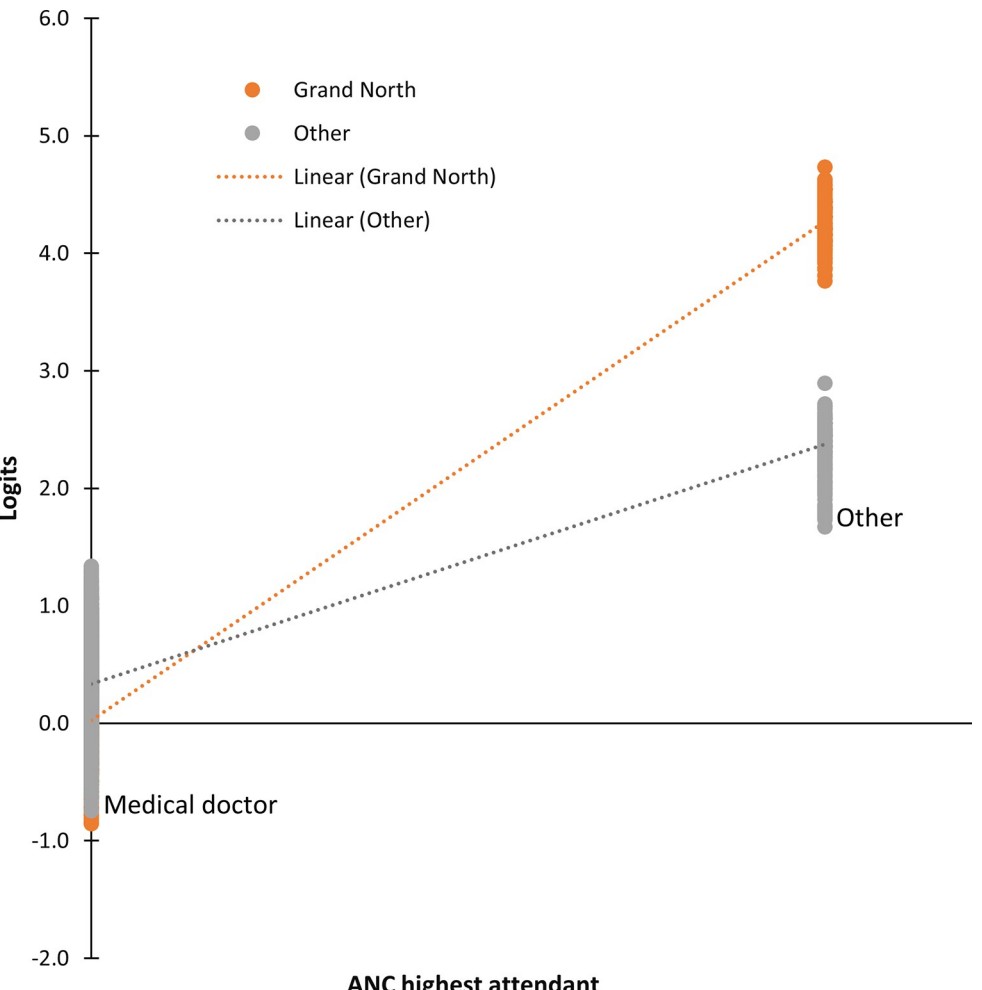

**Fig 3. Cross-level interaction (ANC highest attendant × Sahelian regions) plot.**

found that a high proportion (61.5%) of women of childbearing age did not receive the recommended three or more doses of IPTp-SP. The factors associated with the poor uptake of IPTp-SP included residing in areas of lower malaria endemicity, attending the first ANC after the third month of pregnancy, attending fewer than four ANCs, and not being attended to by a healthcare professional during the prenatal period. In line with the Roll Back Malaria Partnership, Cameroon's Ministry of Health aimed to achieve 80% coverage of the recommended three or more doses of IPTp-SP by 2023 [18]. Although the coverage rate of 38.5% observed in our study represents a substantial improvement from the 12.9% reported in 2011 [10], the country is still facing significant challenges in meeting the desired target. However, a cross-country DHS analytical study conducted in 2013 found a wide variation in the coverage of three or more doses of IPTp-SP, ranging from 0.3% to 43% across sub-Saharan African countries, placing Cameroon among countries with a relatively high IPTp coverage. Countries with lower coverage rates of IPTp-SP typically experience factors such as ambiguous or delayed adoption of relevant policies, lower levels of ANC attendance with a greater proportion of missed opportunities for IPTp-SP administration, and systemic issues with healthcare provision, such as drug stock-outs and high user fees [19–22].

In Cameroon, similar to many other sub-Saharan African countries, the distribution of IPTp-SP is primarily facilitated through the ANC program and has become an integral part of it. Consequently, as demonstrated in our results, women who delay their initial contact with healthcare providers and attend a limited number of subsequent consultations are less likely to receive the recommended IPTp-SP dosage. This finding aligns with previous research suggesting that early initiation of ANC improves the uptake of IPTp-SP [20, 23–25] and that the possibility of completing the recommended doses increases with the number of ANCs attended [10, 11, 26–28]. However, only 47.2% of the women who attended four or more visits in our study actually received the recommended doses, indicating a high frequency of missed opportunities for the administration of IPTp-SP.

Barriers to the uptake of three or more doses of IPTp-SP can be categorized into health system-related factors, ANC attendant-related factors, and maternal-related factors. Our results show that the qualifications of ANC attendants can determine the uptake of three or more doses of IPTp-SP, with medical doctors generally providing better care compared to other attendants. A study conducted in the north-west region of the country reported that up to 35.9% of health providers had not received any training on IPT and that approximately 30% were unaware of the appropriate timing for IPT initiation [9]. These findings are consistent with those of numerous studies demonstrating healthcare providers' inadequate knowledge on IPT [20]. We notice that the effect of the ANC attendant's qualification is modified by region such that women who received no care from a healthcare professional in the Sahelian regions had an even higher risk of poor IPTp-SP uptake compared to those in the Southern regions. The Sahelian regions contain three out of the five administrative regions where women face significant challenges in accessing healthcare services [14]. Moreover, the weaker socio-economic situation of these regions contributes to deficiencies in healthcare delivery. The far north, for instance, only has 8% of all Cameroonian doctors despite holding 18% of the population [29]. These findings underscore the surplus value of assigning additional healthcare professionals to obstetric care in Sahelian regions, as suggested by our cross-level interaction analysis. By enhancing the availability of healthcare providers in underserved regions, we can improve the uptake of IPTp-SP and address disparities in maternal healthcare access.

Previous research also reports several maternal-related factors that contribute to the poor uptake of IPTp. We found no association between maternal age and poor uptake of IPTp-SP. Our findings oppose those of a study conducted in the south-west region of Cameroon, which suggested that, although not statistically significant, younger women were showing a limited interest in IPTp-SP [12]. A Nigerian study found similar results [24]. Hence, the effect of age on the uptake of IPTp-SP is uncertain across studies [20]. As a continuous factor, it may be difficult to sufficiently adjust for age using widespread statistical techniques. The association found by certain studies may be a result of the residual confounding due to the improper handling of the independent variable, such as by classification into a limited number of categories [30], rather than a true association, suggesting a probable treatment negligence issue with younger women.

## Strengths and limitations

The representativeness of our findings was ensured by the use of a large sample size and weighted data from a national-level household survey. The data was collected through standardized procedures using validated questionnaires, ensuring the internal and external validity of the results.

This study, however, has several limitations that should be considered while interpreting the results. First, causation cannot be asserted, as the cross-sectional design of the study does

not allow for a temporal relationship between the independent variables and the outcome of interest. Second, most measures were self-reported, which may be subject to social desirability bias and recall bias considering the fact that some women were asked to recall pregnancy-related events that occurred up to five years before the survey. Third, the study was limited by the available variables in the CMRDHS questionnaire and did not investigate other potentially relevant factors, such as access to healthcare facilities, service provider performance, and women's perceptions and social attitudes toward IPTp. Additionally, some of the data collected, such as wealth and place of residence, reflected the women's conditions at the time of the survey and not at the time of childbirth, which may have caused misclassification of some women. However, the probability is low, with equal chances of a backward and forward shift from the actual category of classification; hence, these conditions should not affect the overall interpretation and implications of our findings. Finally, it is important to note that this study focused on Cameroon and used data from 2018. Therefore, the conclusions drawn from this study may not be generalizable to other countries, and the results may be outdated by the time of publication.

## Conclusion

Despite tremendous progress in the past decades, the coverage of three doses of IPTp-SP in Cameroon remains low. Maternal age, malaria endemicity, place of residence, number of ANCs attended, timing of first ANC, and qualification of ANC attendant (which has a significantly different effect for women residing in the Sahelian regions) were identified as important predictors of IPTp-SP uptake in our study. This finding highlights the importance of early initiation of ANC and consistent attendance for optimal IPTp-SP uptake. The qualification of the ANC attendant also emerged as a significant determinant, emphasizing the necessity for the comprehensive training of and knowledge dissemination among healthcare providers to ensure the appropriate administration of IPTp-SP. However, further studies should be conducted to explore ANC attendant-related factors that could explain the high frequency of missed opportunities for IPTp-SP administration. Policymakers must prioritize the Sahelian regions and address the regional disparities in service availability. Developing targeted strategies that overcome these challenges is essential for improving IPTp-SP uptake and reducing the burden of malaria in pregnancy.

## Acknowledgments

We are grateful to *The DHS Program* team, who granted us access to the 2018 Cameroon Demographic and Health Survey database.

## Author Contributions

**Conceptualization:** Dominique Ken Guimsop, Jerome Ateudjieu.

**Data curation:** Dominique Ken Guimsop, Ange Faustine Kenmogne Talla.

**Formal analysis:** Dominique Ken Guimsop, Ange Faustine Kenmogne Talla.

**Methodology:** Dominique Ken Guimsop, Ange Faustine Kenmogne Talla, Jerome Ateudjieu.

**Project administration:** Dominique Ken Guimsop.

**Supervision:** Jerome Ateudjieu.

**Validation:** Dominique Ken Guimsop, Jerome Ateudjieu.

**Writing – original draft:** Dominique Ken Guimsop, Ange Faustine Kenmogne Talla, Haoua Kodji.

**Writing – review & editing:** Dominique Ken Guimsop, Jerome Ateudjieu.

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
