## [Decision Letter · Decision Letter 0]

13 Dec 2022

PGPH-D-22-01640

Coverage and factor associated to the uptake of Intermittent preventive treatment for malaria during pregnancy (IPTp) in Cameroun in 2018: an analysis from the 2018 Cameroon Demographic Health Survey data

Dear Dr. Guimsop,

Thank you for submitting your manuscript to PLOS Global Public Health. After careful consideration, we feel that it has merit but does not fully meet PLOS Global Public Health’s publication criteria as it currently stands. Therefore, we invite you to submit a revised version of the manuscript that addresses the points raised during the review process.

We look forward to receiving your revised manuscript.

Kind regards,

Orvalho Augusto, MD, MPH

Academic Editor

Journal Requirements:

Additional Editor Comments (if provided):

Reviewers' comments:

Reviewer's Responses to Questions

**Comments to the Author**

1. Does this manuscript meet PLOS Global Public Health’s publication criteria? Is the manuscript technically sound, and do the data support the conclusions? The manuscript must describe methodologically and ethically rigorous research with conclusions that are appropriately drawn based on the data presented.

Reviewer #1: Yes

Reviewer #2: Yes

2. Has the statistical analysis been performed appropriately and rigorously?

Reviewer #1: Yes

Reviewer #2: Yes

3. Have the authors made all data underlying the findings in their manuscript fully available (please refer to the Data Availability Statement at the start of the manuscript PDF file)?

Reviewer #1: Yes

Reviewer #2: No

4. Is the manuscript presented in an intelligible fashion and written in standard English?

Reviewer #1: Yes

Reviewer #2: No

5. Review Comments to the Author

Reviewer #1: This is an interesting paper in an important area.

Comments

Lines 29-30: you mentioned, "Frequencies of mother exposed to less than 3 doses of IPTp-SP were estimated with a 95% CI". However, I went through all the results but did not see where you presented these frequencies with their confidence intervals

Line 32: you did not specify the statistical model used to estimate the crude and adjusted odd ratios, but this should be mentioned in the Abstract's method section.

Line 45: This is the conclusion and cannot be started with the word "Therefore." It would be best if you replaced it with an appropriate term for a conclusion like to sum up, In conclusion, or even to conclude.

Line 47: You did within comparison for the Sahelian region, but it would be more interesting to compare the sahelians to the others regions and not within the same region.

Line 66: can you provide more data on the overall range of the coverage in terms of the minimum and the maximum?

Line 76 – 77: You mentioned that the gap was to understand the limitation of the health care system in achieving its objectives; meanwhile, in this current study, the main targets was to determine the coverage and understand the socio-demographics parameters that affect the uptake of IPTp-SP in Cameroon. In other words, you did not collect any healthcare system parameters in your study. Therefore, there is a need to re-formulate this sentence

Line 89: can you elaborate more on the two-stage sampling process, I mean, the primary sampling unit, the secondary sampling unit, and so on?

Line 95: There is a missing word cause you mentioned "A living child less than at the day the selected household is visited. I think the missing word is five. Cause you mean less than 5 years old?

Lines 110: The definition of the dependent variables is not clear. The dependent variable is the uptake of IPTp-SP and comprises two categories, < 3 doses and ≥ 3 doses. Therefore, there is a need to rewrite this sentence.

Line 122: you mentioned that you used the complex sample's functionality in SPSS due to the structure of your data. Does this account for the cluster sampling aspect of your study? In other words, does this complex sample functionality account for mixed effects models?

Line 128: You used binary logistic regression to analyze your data. I don't think it is appropriate because you used clustered data, so using mixed effects models to account for this cluster effect is more appropriate.

Line 130: You mentioned, "Dependent that had an association with the outcome." Dependent, outcome, and response variables are the same. Therefore, there is a need to correct this. I think you mean independent, which had an association with the outcome.

Line 157: Table 1: I prefer the Mother's age (years) instead of Age groups (years). Also, it is better to present the proportions row-wise instead of column-wise. In addition, you should compare the proportions between levels instead of within levels.

Lines 169: You started describing the data without indicating in which Tables we can find these data. In the first sentence of this paragraph, you should introduce the Table and then follow with the data description.

Line 180: You should provide a between-regions comparison and not within regions.

Line 184: You should first introduce the tables, and the proportion should be presented row-wise instead of column-wise as you did.

Line 186: You mentioned parity below 6, but in the Table, we only have parity ≤ 5.

Line 217: You omitted to add Table 3 before the title "Determinants of the uptake ....

Reviewer #2: Comments

In this paper the authors describe the coverage and factors associated with the uptake of intermittent preventive treatment for malaria in pregnant women in Cameroon. This is an important topic because the available body of evidence has shown that in malaria endemic settings the uptake of IPT-SP of � 3 significantly improves the pregnancy outcomes. This manuscripts will help to inform the policy makers in Cameroon on the prevailing gaps in the uptake of IPT-SP � 3 doses, and therefore, assist in making decisions.

Major comment

The authors should find a native English speaker to assist correcting the language.

Minor comments

Line 21: Change …..increasing mother and children mortality…….to…..increasing pregnant women/mothers and children mortality…..

Line 23: WHO is mentioned for the first time, thus it should be written in full followed by the abbreviations in brackets, thereafter, you can continue using the abbreviations.

Line 24: Present study aimed…and not…aims.

Line 27: Change…administered in face to face…to….administered face to face…or ,..administered directly.

Line 28:….one child under 5…what is the unit of measurement, years or months?

Line 30: …..A case-control design….Can you explain how this was a case control design study?

Line 43: ……were preventing factors for poor uptake of IPTp-SP 3+. This statement is not clear, change it to …were factors associated with inadequate uptake of ……. In addition change….highest level….to….high/higher level…..

Line 44:…..not found associated to……not found to be associated with…..

Line 45: The conclusion is not clear, re-write to increase its clarity.

Line 65: Despite implemented…..to….Despite the implementation of…..

Line 68: …..factors as level…..to….factors such as /or including…….

Line 70: The sentence is not clear, re-write to increase its clarity…e.g.,…the Ministry of Health adopted IPTp using chloroquine since 2002….

Line 71: ANC….write in full, followed by the abbreviations in brackets, thereafter, you can continue using the abbreviation.

Line 72: The sentence is not clear, rephrase.

Line 95: ……biological mothers of a living child aged less than….less than what? Also, ……household is visited…….change to…..household was visited.

Line 100: ….a under 5 years old….to…an under 5 years old….

Line 102: …..and those did not have…..to….and those who did not have…..

Line 111: …..stay in the place of residence……change to……residents of the catchment area…..

Line 112:Why have both “woman’s highest education level” and “literacy”? These two assess the same thing.

Line 115: ANC highest attendant….this is not clear. I believe you meant the cadres of the attendants i.e., nurse, doctor, etc. Find a good way to express it to increase clarity.

Line 130-131: Dependents that…..were selected for multivariate logistic regression model. This sentence is not clear, rephrase to increase clarity.

Line 151:……instead of (48.1%, n=2657)…..change to…..48.1% (2657/5528). Change throughout the results section.

Line 151-52:….with almost half of aged…to….with almost half of them aged…., also add a comma between “old” and “and”.

Line 152: ….., and 44.5% having….to…and 44.5% of them having…..

Line 153: About half (52.6% (2911/5528)…..this is more than half, not about half. Also change….and they seem to….to……and they seemed to….

Table 1: -Why the age group of 40-49 has a range of 9 years while all other age groups have a range of 4 years?

-Stay in the place of residence…to.. reside/or resident of the catchment area.

-Highest education level…..to level of education.

-Illiterate is part of the level of education, so it should be combined with no formal education.

-Current occupation…….I was expecting to see different groups of occupation i.e., peasants, petty traders, government employee etc.

Table 2: ANC highest attendant…….change to …Cadre of the ANC attendant…

Line 167: Subheading….Sociodemographic characteristics of participants and uptake of �3 doses of IPTp-SP……refers to the same things as the subheading…..Determinants of the uptake of less than 3 doses of IPTp-SP. These two sections should be combined and streamlined to avoid repetition.

Line 227: Since 2004 in Cameroon……to…In Cameroon, since 2004 IPTp-SP…..

Line 275: ….service overwhelming……to….health system overwhelmed…..

6. PLOS authors have the option to publish the peer review history of their article (what does this mean?). If published, this will include your full peer review and any attached files.

**Do you want your identity to be public for this peer review?** For information about this choice, including consent withdrawal, please see our Privacy Policy.

Reviewer #1: No

Reviewer #2: **Yes: **Richard Mwaiswelo

---

## [Decision Letter · Decision Letter 1]

23 Feb 2023

PGPH-D-22-01640R1

Coverage and factor associated to the uptake of Intermittent preventive treatment for malaria during pregnancy (IPTp) in Cameroun in 2018: an analysis from the 2018 Cameroon Demographic Health Survey data

Dear Dr. Guimsop,

Thank you for submitting your manuscript to PLOS Global Public Health. After careful consideration, we feel that it has merit but does not fully meet PLOS Global Public Health’s publication criteria as it currently stands. Therefore, we invite you to submit a revised version of the manuscript that addresses the points raised during the review process.

We look forward to receiving your revised manuscript.

Kind regards,

Orvalho Augusto, MD, MPH

Academic Editor

Journal Requirements:

Additional Editor Comments (if provided):

Reviewers' comments:

Reviewer's Responses to Questions

**Comments to the Author**

1. If the authors have adequately addressed your comments raised in a previous round of review and you feel that this manuscript is now acceptable for publication, you may indicate that here to bypass the “Comments to the Author” section, enter your conflict of interest statement in the “Confidential to Editor” section, and submit your "Accept" recommendation.

Reviewer #3: (No Response)

Reviewer #4: (No Response)

2. Does this manuscript meet PLOS Global Public Health’s publication criteria? Is the manuscript technically sound, and do the data support the conclusions? The manuscript must describe methodologically and ethically rigorous research with conclusions that are appropriately drawn based on the data presented.

Reviewer #3: Yes

Reviewer #4: Yes

3. Has the statistical analysis been performed appropriately and rigorously?

Reviewer #3: I don't know

Reviewer #4: Yes

4. Have the authors made all data underlying the findings in their manuscript fully available (please refer to the Data Availability Statement at the start of the manuscript PDF file)?

Reviewer #3: Yes

Reviewer #4: No

5. Is the manuscript presented in an intelligible fashion and written in standard English?

Reviewer #3: Yes

Reviewer #4: Yes

6. Review Comments to the Author

Reviewer #3: General

1. Title: ‘Coverage and factor associated to the uptake of Intermittent preventive treatment for malaria during pregnancy (IPTp) in Cameroun in 2018: an analysis from the 2018 Cameroon Demographic Health Survey data’ could be revised to ‘Factors associated with the uptake of Intermittent preventive treatment for malaria during pregnancy (IPTp) in Cameroun in 2018: an analysis from the 2018 Cameroon Demographic Health Survey data’ as Coverage is implied in the ‘uptake’.

2. Generally, it would be good to have a fluently speaking English health personnel edit the manuscript for general improvement in the expression of the language in the manuscript as there are some few errors in some of the sentences. For example, the first sentence of the Abstract in lines 20-21 ‘Malaria in pregnancy is a major public health issue, contributing to a significant increase in pregnant women and children mortality and morbidity’ could read as follows: ‘Malaria in pregnancy is a major public health issue, contributing to a significant increase in maternal and child mortality and morbidity’. And in lines 27-28, ‘Data were collected using a questionnaire administered in face to face to mothers with at least one child under 5, selected using a 2-stage stratified sampling process.’ Could read ‘Data were collected using a questionnaire administered face to face to mothers with at least one child under 5, selected using a 2-stage stratified sampling process.’ I recommend a thorough reading through the manuscript to correct minor typographical and grammatical errors.

Abstract

3. Authors mention in line 30 that ‘a case-control design was used to assess…’. Could the study design be an analytical cross-sectional design instead? Demographic and Health surveys are cross-sectional studies.

Introduction

4. Line 64: ‘once month apart’ should read ‘one month apart’

5. Line 71: ‘u’ is missing from ‘sing’ to read ‘using’ chloroquine… Line 78, antenatal is spelt wrongly at the beginning of the sentence, t is also missing from ‘that’.

6. Line 80: poor uptake has been referred to as IPTp-SP ≥ 3 doses. This should be corrected.

Study population

7. Do Authors mean biological mothers of children less than five years who were between the ages of 15-49 years? The statement in lines 101 – 103 is not clear.

Data management

8. Data grooming is not the same as data cleaning. Did Authors mean data cleaning in line 126?

Statistical analysis

9. Line 137: ‘our’ should be replaced by ‘the’. Better still, the outcome of interest could be repeated here.

10. Line 143: What do Authors mean by ‘interactions among our dependents were explored’? Are they referring to (in)dependent variables?

Results

11. Line 167: Is that sentence a heading?

12. Line 173: The caption of Figure 2 needs modification. Proportion of uptake does not read well as ‘uptake’ means the proportion of women who have taken a specific number of IPTp-SP doses.

13. Lines 180-183: The interpretations Authors are giving to the figures here are not correct. Authors should interpret the results carefully. Secondly, the degrees of freedom that are reported in Table 1 are very wide; Authors need to state cautiously what the findings are at this stage.

14. Again, the interpretation of results in lines 201 to 203 is faulty. Authors should kindly look carefully at these. For example, saying that ‘a greater proportion of women having a poor uptake of IPTp-SP among those who received no care (96.6%)’ is not the same as saying that ‘15.3% of the respondents did not receive any ANC with 96.6% of these having a poor uptake of IPTp-SP.’

15. Table 3: Women’s age category should be 15-25 years according to previous categorization in table 1 and not 0-25 years.

16. Line 220: Could Authors consider stating the positive finding instead of the double negative as stated in the sentence? This would state the finding more clearly.

Discussion

17. Lines 287 to 291: These are very valid reasons that have been written in this paragraph. Could Authors clarify/simplify the sentences in these lines and possibly add some references? Does ‘increase in drug shortage’ mean SP stockouts? What does ‘an overwhelmed health system from the elevated demand’ mean? And does ‘dodging of preventive doses because of recent treatment received for frequent infections’ mean ‘treatment of malaria infections among the pregnant women because they report with signs and symptoms of malaria during ANC rather than giving IPTp presumptively? The policy demands that pregnant women who show signs and symptoms of malaria get tested and those positive treated using ACTs instead of giving IPTp hence the potential reduction in IPTp-SP. This could be elaborated in the discussion.

Reviewer #4: Manuscript PGPH-D-22-01640: Coverage and factor associated to the uptake of Intermittent preventive treatment for malaria during pregnancy (IPTp) in Cameroun in 2018: an analysis from the 2018 Cameroon Demographic Health Survey data

General comments - The manuscript addresses an important of prevention of malaria in pregnancy, using existing large population-level data; the national demographic health survey.

Specific Comments

Abstract – The abstract is well written and summarizes the rationale for the study, methodology, key findings, and conclusions.

Introduction - The authors have cited relevant studies and provided good background, focusing on the main focus of the manuscript. It flows well.

Materials and Methods –General: Given that this paper is based on secondary analysis of data collected for another purpose, the details provided on how the original data were cleaned, transformed and manipulated to fit the objectives of the present analysis are missing. Expand description on what figure 1 is showing as well, especially explaining the exclusions of the “unweighted”. I would recommend that authors should review some DHS-based papers to see how they have structured the methods sections.

- Lines 109-111: (a) the dependent variable 110 was the number of women who had received less than three doses of IPTp-SP (Poor uptake of 111 ≥3 doses of IPTp-SP). The inclusion of the words in bracket may confuse some readers; consider leaving it out.

- Line 111: The use of the word “grooming” seems inappropriate; consider changing to cleaning, preparation for analysis, etc.

Results – Findings presented well.

- Tables 1&2: The presentation of “Rao-Scott-F(df1, df2)” and “p-value” columns for categorical variables is a bit confusing; it seems to be linked to the first row of the variable, while it applies to the entire variable. Merge the spaces.

- “The 3rd table is not properly labeled as “Table 3”; correct this. In the same table, consider using “REF/Reference” instead of the value “1” in the OR column.

Discussion – This section can be improved/ strengthen by:

- Avoid repeating the presentation of results especially the 1st paragraph.

- Include a few sentences either at the beginning or at the end describing the strengths of this study, such as use of a large set of database that is nationally representative, etc.

- Include the limitations of the study; this information is missing.

References – Relevant and recent.

Other comments – Minor language editing required to improve clarity and conform to English grammar.

7. PLOS authors have the option to publish the peer review history of their article (what does this mean?). If published, this will include your full peer review and any attached files.

**Do you want your identity to be public for this peer review?** For information about this choice, including consent withdrawal, please see our Privacy Policy.

Reviewer #3: No

Reviewer #4: **Yes: **Mark M Kabue

---

## [Decision Letter · Decision Letter 2]

20 Jun 2023

PGPH-D-22-01640R2

Factors associated with the uptake of Intermittent preventive treatment for malaria during pregnancy (IPTp) in Cameroon: an analysis of data from the 2018 Cameroon Demographic Health Survey

Dear Dr. Guimsop,

Thank you for submitting your manuscript to PLOS Global Public Health. After careful consideration, we feel that it has merit but does not fully meet PLOS Global Public Health’s publication criteria as it currently stands. Therefore, we invite you to submit a revised version of the manuscript that addresses the points raised during the review process.

We look forward to receiving your revised manuscript.

Kind regards,

Orvalho Augusto, MD, MPH

Academic Editor

Journal Requirements:

Additional Editor Comments (if provided):

IPTp is an important strategy for malaria prevention in pregnancy. The authors analyzed recent Cameroon DHS data to identify the level of low uptake of IPTp in Cameroon and try to identify important factors. There are a few shortcomings in the report.

1. Somewhere in the abstract, it is said that the author conducted a multilevel analysis. However, we have no details of such a procedure in the methods, neither the tables (eg table 3) with associations suggest such a procedure has ever been used here.

- Please add more details in the statistical analysis of what was done to perform this analysis.

2. The abstract is too long. Please add the subsections (background, methods, results, and conclusions to help the reader follow).

Reviewers' comments:

Reviewer's Responses to Questions

**Comments to the Author**

1. If the authors have adequately addressed your comments raised in a previous round of review and you feel that this manuscript is now acceptable for publication, you may indicate that here to bypass the “Comments to the Author” section, enter your conflict of interest statement in the “Confidential to Editor” section, and submit your "Accept" recommendation.

Reviewer #5: (No Response)

2. Does this manuscript meet PLOS Global Public Health’s publication criteria? Is the manuscript technically sound, and do the data support the conclusions? The manuscript must describe methodologically and ethically rigorous research with conclusions that are appropriately drawn based on the data presented.

Reviewer #5: Partly

3. Has the statistical analysis been performed appropriately and rigorously?

Reviewer #5: No

4. Have the authors made all data underlying the findings in their manuscript fully available (please refer to the Data Availability Statement at the start of the manuscript PDF file)?

Reviewer #5: Yes

5. Is the manuscript presented in an intelligible fashion and written in standard English?

Reviewer #5: No

6. Review Comments to the Author

Reviewer #5: 1. According to the manuscript, the objective was to determine factors contributing to the low uptake of IPTp. But the authors presented factors for both high and low uptake of IPTp in results, discussion and conclusion sections; this may confuse readers of the article. I recommend the authors to put their focus on the factors for low uptake of IPTp.

2. The authors should consider revising the analysis (especially the logistic regression). In most cases, the multivariate logistic regression analysis includes independent variables that have attained a certain level of significance (usually a p-value of less than 0.2) during the univariate logistic regression analysis. This will enable authors to (1) consider only significant variables and (2) adjust for confounders.

3. Authors should rewrite the references appropriately and consistently both in text (using brackets instead of parentheses) and the reference list, e.g., reference number 2. Authors should adhere to the journal guidelines for writing references (Vancouver style).

4. To improve the manuscript's language, the authors should send it to a professional English editor.

7. PLOS authors have the option to publish the peer review history of their article (what does this mean?). If published, this will include your full peer review and any attached files.

**Do you want your identity to be public for this peer review?** For information about this choice, including consent withdrawal, please see our Privacy Policy.

Reviewer #5: No

---

## [Decision Letter · Decision Letter 3]

25 Sep 2023

PGPH-D-22-01640R3

Factors associated with the uptake of Intermittent preventive treatment for malaria during pregnancy (IPTp) in Cameroon: an analysis of data from the 2018 Cameroon demographic and health survey

Dear Dr. Guimsop,

Thank you for submitting your manuscript to PLOS Global Public Health. After careful consideration, we feel that it has merit but does not fully meet PLOS Global Public Health’s publication criteria as it currently stands. Therefore, we invite you to submit a revised version of the manuscript that addresses the points raised during the review process.

We look forward to receiving your revised manuscript.

Kind regards,

Orvalho Augusto, MD, MPH

Academic Editor

Journal Requirements:

Additional Editor Comments (if provided):

Reviewers' comments:

Reviewer's Responses to Questions

**Comments to the Author**

1. If the authors have adequately addressed your comments raised in a previous round of review and you feel that this manuscript is now acceptable for publication, you may indicate that here to bypass the “Comments to the Author” section, enter your conflict of interest statement in the “Confidential to Editor” section, and submit your "Accept" recommendation.

Reviewer #5: All comments have been addressed

Reviewer #6: All comments have been addressed

2. Does this manuscript meet PLOS Global Public Health’s publication criteria? Is the manuscript technically sound, and do the data support the conclusions? The manuscript must describe methodologically and ethically rigorous research with conclusions that are appropriately drawn based on the data presented.

Reviewer #5: Yes

Reviewer #6: Yes

3. Has the statistical analysis been performed appropriately and rigorously?

Reviewer #5: Yes

Reviewer #6: No

4. Have the authors made all data underlying the findings in their manuscript fully available (please refer to the Data Availability Statement at the start of the manuscript PDF file)?

Reviewer #5: Yes

Reviewer #6: Yes

5. Is the manuscript presented in an intelligible fashion and written in standard English?

Reviewer #5: No

Reviewer #6: Yes

6. Review Comments to the Author

Reviewer #5: Title for table 1, what are the environmental characteristics presented in the table? Revise the title using appropriate terminology.

Revise the titles for subsections in the result section, for example the first subsection may be "Demographic Characteristics of Study population".

Reviewer #6: Title: Factors associated with the uptake of Intermittent preventive treatment for malaria during pregnancy (IPTp) in Cameroon: an analysis of data from the 2018 Cameroon demographic and health survey.

GENERAL COMMENTS:

The manuscript presents findings from a secondary data analysis of the 2018 Cameroon Demographic and Health Survey, which involved 5409 (unweighted) childbearing women from across Cameroon, all of whom had at least one child under five years old at the time of the study. This research addresses a crucial aspect of public health by examining the distribution and determinants of poor uptake of Intermittent Preventive Treatment for malaria during pregnancy (IPTp-SP), particularly in cases where pregnant women received fewer than three doses. While the manuscript is well-written, there is room for improvement in certain areas.

ABSTRACT, TITLE

The abstract is well-structured, but I have a question regarding the focus of conclusion section. It seems the authors are emphasizing the determinants of optimal uptake of IPTp-SP (≥ 3 doses) rather than the factors associated with poor IPTp-SP uptake, which aligns better with the study's aim. I suggest that they maintain consistency with the study's objectives in presenting the conclusion.

Title is relevant and informative and the aim is clear

MAIN MANUSCRIPT CONTENTS

1. Background

In lines 73-76, the authors have mentioned the prevalence range of optimal IPTp-SP uptake in relation to socio-demographic characteristics, ANC (Antenatal Care) utilization, and study design in Cameroon. They also pointed out that there have been few studies conducted in Cameroon that explored the association of socio-demographic factors and “antennal parameter” (what is the antennal?) with IPTp-SP uptake (as seen in lines 77-78). Given this information, I am not entirely convinced about the novel contribution of this present study to the literature on IPTp-SP uptake in Cameroon. The authors have not thoroughly critiqued previous studies conducted in Cameroon to establish the specific gap that this study aims to address.

2. Methods

Statistical analysis

In line 161, the authors have stated that all independent variables were dichotomized to assess their association with IPTp uptake. It is important to note that logistic regression can be applied to independent variables with various levels of measurement, not just dichotomous variables. Dichotomizing variables is a well-known practice, but it can lead to a loss of information and potentially introduce residual confounding. Therefore, I have reservations about the rationale behind this extensive dichotomization.

Statistical model used

In lines 160-161, the manuscript mentions the use of bivariate logistic regression. It is not clear whether this analysis is within a multilevel modelling (MLM) framework or standard logistic regression. If it's standard logistic regression, an explanation for this choice would be beneficial.

Additionally, in line 172, it is indicated that MLM was applied for each covariate. The rationale for employing standard regression in the initial stage and MLM in the second stage should be clarified.

Furthermore, in lines 188-189, the manuscript states that the analysis accounted for the survey design. If multilevel modelling analysis incorporated weights, it would be valuable to provide a brief description of the methods used in this regard.

In line 192, the manuscript mentions the fitting of interaction terms. It would be beneficial to clarify whether these interaction terms were pre-planned as part of the study's hypothesis and design or if they were determined through a trial-and-error process during the analysis. Understanding the rationale behind the inclusion of interaction terms can provide useful insight.

3. Results

In the methods section, the authors state that all independent variables were dichotomized (line 161). However, upon reviewing Table 1, it is evident that not all variables are dichotomized. The methods section should be revised for accuracy and consistency.

Additionally, in Table 1, it would be beneficial for the authors to include a footnote indicating that the sample sizes presented are weighted, as this is important information for the readers to understand.

Furthermore, in line 233, the authors claim a statistically significant difference between higher education and secondary education, even though there are four levels of education mentioned. It is unclear which specific level(s) of education are driving this statistical significance. In my opinion, it is difficult for the authors to specify the particular education levels for which a statistically significant difference was found. The way conclusions are drawn involving more than two levels of variable measures should be revised in this manuscript.

Does the Table 3 include the MLM bivariable logistic regression or the standard logistic regression?

4. Discussion

I recommend that the statements found in lines 292-299 be incorporated into the 'strengths of the study' section, as they seem to highlight specific strengths or advantages of the research.

Results in lines 342-343 should be interpreted with caution because the p-value is weak as evidence with 95% confidence.

7. PLOS authors have the option to publish the peer review history of their article (what does this mean?). If published, this will include your full peer review and any attached files.

**Do you want your identity to be public for this peer review?** For information about this choice, including consent withdrawal, please see our Privacy Policy.

Reviewer #5: No

Reviewer #6: **Yes: **Steven Chifundo Azizi

---

## [Decision Letter · Decision Letter 4]

15 Jan 2024

PGPH-D-22-01640R4

Factors associated with the uptake of Intermittent preventive treatment for malaria during pregnancy (IPTp) in Cameroun: an analysis of data from the 2018 Cameroon demographic and health Survey

Dear Dr. Guimsop,

Thank you for submitting your manuscript to PLOS Global Public Health. After careful consideration, we feel that it has merit but does not fully meet PLOS Global Public Health’s publication criteria as it currently stands. Therefore, we invite you to submit a revised version of the manuscript that addresses the points raised during the review process.

We look forward to receiving your revised manuscript.

Kind regards,

David Musoke, PhD

Academic Editor

Journal Requirements:

Additional Editor Comments (if provided):

Thank you for sharing your revised manuscript. The reviewers have identified some areas that still need to be addressed.

Reviewers' comments:

Reviewer's Responses to Questions

**Comments to the Author**

1. If the authors have adequately addressed your comments raised in a previous round of review and you feel that this manuscript is now acceptable for publication, you may indicate that here to bypass the “Comments to the Author” section, enter your conflict of interest statement in the “Confidential to Editor” section, and submit your "Accept" recommendation.

Reviewer #5: (No Response)

Reviewer #6: (No Response)

2. Does this manuscript meet PLOS Global Public Health’s publication criteria? Is the manuscript technically sound, and do the data support the conclusions? The manuscript must describe methodologically and ethically rigorous research with conclusions that are appropriately drawn based on the data presented.

Reviewer #5: Yes

Reviewer #6: Yes

3. Has the statistical analysis been performed appropriately and rigorously?

Reviewer #5: Yes

Reviewer #6: Yes

4. Have the authors made all data underlying the findings in their manuscript fully available (please refer to the Data Availability Statement at the start of the manuscript PDF file)?

Reviewer #5: Yes

Reviewer #6: No

5. Is the manuscript presented in an intelligible fashion and written in standard English?

Reviewer #5: Yes

Reviewer #6: Yes

6. Review Comments to the Author

Reviewer #5: Congratulation to the authors, there is great improvement from previous versions.

Line 278 – 282: “This indicates ......... Specifically, ......... “. These statements should be moved to discussion section.

I believe there is still a need for improvement in terms of language. Hence, I recommend that an English editor revise the manuscript.

Reviewer #6: ABSTRACT, TITLE

The abstract is well-structured, but I have a question regarding the focus of conclusion section. It seems the authors are emphasizing the determinants of optimal uptake of IPTp-SP (≥ 3 doses) rather than the factors associated with poor IPTp-SP uptake, which aligns better with the study's aim. I suggest that they maintain consistency with the study's objectives in presenting the conclusion.

Your response on the above suggestion shows that you did not understand what I meant. It is not about the mere word “determinant” but your aim and conclusion are not matching. You aimed to investigate the distribution and factors associated with poor uptake of IPTp-SP… but your conclusion is highlighting the optimal uptake of IPTp-SP.

MAIN MANUSCRIPT CONTENTS

1. Background

In lines 73-76, the authors have mentioned the prevalence range of optimal IPTp-SP uptake in relation to socio-demographic characteristics, ANC (Antenatal Care) utilization, and study design in Cameroon. They also pointed out that there have been few studies conducted in Cameroon that explored the association of socio-demographic factors and “antennal parameter” (what is the antennal?) with IPTp-SP uptake (as seen in lines 77-78). Given this information, I am not entirely convinced about the novel contribution of this present study to the literature on IPTp-SP uptake in Cameroon. The authors have not thoroughly critiqued previous studies conducted in Cameroon to establish the specific gap that this study aims to address.

I remain unconvinced by the new justification. You have cited only two studies (DHS and Diengou et al, 2020) as your evidence, yet there are numerous studies conducted in Cameroon on this subject. Perhaps it would have been more accurate to mention the limited literature available on this topic from a nationally representative sample, which the Demographic and Health Survey provides.

3. Results

In the methods section, the authors state that all independent variables were dichotomized (line 161). However, upon reviewing Table 1, it is evident that not all variables are dichotomized. The methods section should be revised for accuracy and consistency.

Based on your response, why can you not add the statement that you used to respond to my query? I suggest that you include the statement that you used to respond to my query.

Additionally, in Table 1, it would be beneficial for the authors to include a footnote indicating that the sample sizes presented are weighted, as this is important information for the readers to understand.

Information presented in Tables should provide comprehensive details without relying on references to other parts of the document. It is concerning to find a table containing information that depends on definitions found in specific sections of the document. Tables should be self-contained and independent. I am not impressed with your response.

7. PLOS authors have the option to publish the peer review history of their article (what does this mean?). If published, this will include your full peer review and any attached files.

**Do you want your identity to be public for this peer review?** For information about this choice, including consent withdrawal, please see our Privacy Policy.

Reviewer #5: No

Reviewer #6: **Yes: **Steven Chifundo Azizi

---

## [Decision Letter · Decision Letter 5]

4 Mar 2024

Factors associated with the uptake of intermittent preventive treatment for malaria during pregnancy in Cameroon: An analysis of data from the 2018 Cameroon Demographic and Health Survey

PGPH-D-22-01640R5

Dear Dr. Guimsop,

We are pleased to inform you that your manuscript 'Factors associated with the uptake of intermittent preventive treatment for malaria during pregnancy in Cameroon: An analysis of data from the 2018 Cameroon Demographic and Health Survey' has been provisionally accepted for publication in PLOS Global Public Health.

Best regards,

David Musoke, PhD

Academic Editor

Reviewer Comments (if any, and for reference):

Reviewer's Responses to Questions

**Comments to the Author**

1. If the authors have adequately addressed your comments raised in a previous round of review and you feel that this manuscript is now acceptable for publication, you may indicate that here to bypass the “Comments to the Author” section, enter your conflict of interest statement in the “Confidential to Editor” section, and submit your "Accept" recommendation.

Reviewer #6: All comments have been addressed

2. Does this manuscript meet PLOS Global Public Health’s publication criteria? Is the manuscript technically sound, and do the data support the conclusions? The manuscript must describe methodologically and ethically rigorous research with conclusions that are appropriately drawn based on the data presented.

Reviewer #6: Yes

3. Has the statistical analysis been performed appropriately and rigorously?

Reviewer #6: Yes

4. Have the authors made all data underlying the findings in their manuscript fully available (please refer to the Data Availability Statement at the start of the manuscript PDF file)?

Reviewer #6: No

5. Is the manuscript presented in an intelligible fashion and written in standard English?

Reviewer #6: Yes

6. Review Comments to the Author

Reviewer #6: (No Response)

7. PLOS authors have the option to publish the peer review history of their article (what does this mean?). If published, this will include your full peer review and any attached files.

**Do you want your identity to be public for this peer review?** For information about this choice, including consent withdrawal, please see our Privacy Policy.

Reviewer #6: **Yes: **Steven Chifundo Azizi
